# Religious leaders' perceptions of the identification and referral of people with mental health problems in a Peruvian city

**Julio Cjuno**[1], **Jessica Hanae Zafra-Tanaka**[2], **Teresa del Pilar García García**[3], **Alvaro Taype-Rondan**[4,5]*

1 Universidad Cesar Vallejo, Escuela de Medicina, Piura, Peru, 2 Universidad Científica del Sur, Escuela de Medicina, Lima, Peru, 3 Universidad Católica Los Ángeles de Chimbote, Escuela Profesional de Psicología, Chimbote, Peru, 4 Universidad San Ignacio de Loyola, Unidad de Investigación para la Generación y Síntesis de Evidencias en Salud, Lima, Peru, 5 EviSalud–Evidencias en Salud, Lima, Peru

* alvaro.taype.r@gmail.com

**Data Availability Statement:** All relevant data are within the manuscript and its Supporting Information files.

## Abstract

### Introduction

Religious leaders have the potential to play a significant role in the identification and referral of individuals with mental health problems.

### Objective

This study sought to understand the perceptions of religious leaders in regards to identifying and referring parishioners with mental health issues to healthcare professionals, in Chimbote, Peru.

### Methods

We performed a cross-sectional study that covered religious leaders of different religious groups in Chimbote. The leaders completed a survey that assessed their characteristics, past experiences of detecting and referring those with mental health problems to healthcare professionals, and perceptions of four clinical cases (for which we used the Clergy's Perception of Mental Illness Survey instrument).

### Results

We included 109 religious' leaders of four religious groups (11 Catholics, 70 Evangelicals, 21 Mormons, and 7 Adventists). Of these, 50.5% had received at least one request for help with mental health issues from a parishioner in the previous month, over 85% expressed a desire for training in identifying mental health problems, and 22–30% reported receiving any training. While the majority of leaders were able to correctly identify cases of depression, alcohol dependence, and drug problems, only 62% correctly classified a case of schizophrenia. Despite this, 80% stated that they would refer their parishioners to healthcare professionals.

**Funding:** The author(s) received no specific funding for this work.

**Competing interests:** The authors have declared that no competing interests exist.

## Conclusion

Parishioners tend to consult their religious leaders regarding their mental health and approximately 80% stated they would refer such cases to a healthcare professional. However, less than one-third of the leaders had received training to detect mental health problems. These results suggest that there is a need for training programs to improve the ability of religious leaders to identify and refer individuals with mental health issues.

## Introduction

Mental health problems are a significant global health issue, accounting for 7% of disability-adjusted life years (DALYs) worldwide in 2016 [1]. Despite the burden of these disorders, many individuals with mental health issues face barriers in accessing healthcare services, including long distances to health centers, lack of awareness about their condition, mistrust in healthcare professionals, and stigma [2]. As a result, people with mental health problems may seek help from alternative sources [3].

In the United States, religious leaders such as priests and pastors have reported receiving inquiries about mental health issues from their parishioners [4]. This phenomenon may be attributed to their unique position of authority and their close cultural and geographical proximity to their congregants [5]. In fact, research has shown that individuals with higher levels of religiosity are more likely to consult their religious leaders rather than healthcare professionals when experiencing mental health problems [6].

This reliance on religious leaders underscores the potential for collaboration between healthcare services and religious institutions to equip the latter with the skills needed to recognize and refer parishioners with suspected mental health issues to appropriate healthcare services. Studies in the United States [4,7] and Nigeria [8] have found that although religious leaders often receive inquiries about mental health problems from their parishioners, they may not always possess the necessary expertise to accurately recognize these issues and often wish to receive training to do so.

In Latin America, it is estimated that 90% of the population identifies as Christian [9,10]. In Peru specifically, 94.9% of people over 12 years of age have reported that they are religious [11]. As in other settings [4,8], it is likely that parishioners in Latin America seek help from their religious leaders for managing mental health problems. Examining this dynamic could provide an opportunity to identify people with mental health problems and refer them to the healthcare system.

Therefore, this study's objective was to understand the perceptions of religious leaders in regards to identifying and referring parishioners with mental health issues to healthcare professionals, in Chimbote, Peru.

## Methods

### Research design, context, and participants

A cross-sectional design was used and the study was conducted in 2019 in the city of Chimbote. The population of interest were all the religious leaders of the main religious group in Chimbote.

Chimbote is located in the region of Ancash, on the Pacific coast of Peru. It had an estimated population of 206,213 in 2018, with a majority of residents working in the fishery and metallurgical industries [12].

In 2017, a survey found that 91.6% of the population of Chimbote identified as following a religion, with 66.6% being Catholics, 19.7% Evangelicals, 1.4% Jehovah Witnesses, 1.3% Adventists, 0.7% Mormons, and 1.5% belonging to other denominations [11].

Accordingly, we included religious leaders from the most commonly represented religious groups in Chimbote: Catholicism, Evangelicalism, Jehovah's Witnesses, Seventh-day Adventism, and the Church of Jesus Christ of Latter-day Saints (Mormons). The organizational structure and characteristics of each religious group are summarized in Table 1.

## Procedures

The present study was conducted in Chimbote, Peru from June to August 2019. Data collection was carried out by two healthcare professionals who were trained by the principal researcher to use the research instrument for a total of 12 hours over the course of three days.

The interviewers visited the main places of worship for the most prevalent religious groups in Chimbote and sought to obtain contact information for their primary religious leaders. Upon locating these individuals, the interviewers explained the study, asked for their authorization to conduct it, and requested them a list of active religious leaders of their religious group in Chimbote and their contact information.

The primary religious leaders of the Catholic Church, Evangelical Church, Jehovah Witnesses, Seventh-day Adventist Church, and Church of Jesus Christ of Latter-day Saints agreed to participate and provided lists of their respective religious leaders. When contact information for these leaders was not available, the interviewers visited their places of worship to make contact in person.

Once informed consent was obtained from a religious leader, the interviewers conducted an interview in a private setting and administered a self-report questionnaire. Data collection continued until saturation was reached, defined as the point at which all religious leaders of each religious group had been invited to participate or no new leaders of a particular religious group could be found after searching them in their places of worship for three consecutive days.

The data were subsequently digitalized in a Microsoft Excel 2016 database and checked for accuracy by another researcher. Later, it was export to STATA version 15 for statistical analysis.

**Table 1. Characteristics of four of the main religious groups in Chimbote, Peru, 2019.**

| Characteristics | Catholic Church (Catholics) | Evangelical Church (Evangelicals) | Church of Jesus Christ of Latter-day Saints (Mormons) | Seventh-day Adventist Church (Adventists) |
|---|---|---|---|---|
| Denomination of their religious leaders | Priest | Shepherd | Bishop | Shepherd |
| Highest organization level in Chimbote | Diocese of Chimbote | Association of Pastors of Chimbote | Chimbote Stake | Central Eastern Peruvian Mission |
| Number of physical places where religious services are performed (such as parishes) in Chimbote | 18 | 150 | 6 | 35 |
| Number of leaders | 18 | 150 | 21 | 7 |
| Central meeting day | Sundays | Sundays | Sundays | Saturdays |
| Foundations of each religious group | Creed-centered | Salvation by grace and the death of Jesus | Teachings of Jesus Christ | The second coming of the Christ |

## Instrument and variables

The data collection instrument used in this study consisted of three sections. The first section included general questions on demographic characteristics such as religious group, sex, age, country of birth, level of education, duration and location of theology studies, and years in a leadership position.

The second section contained questions on mental health issues such as depression, anxiety, schizophrenia, psychosis, and substance abuse. The participants were asked about their past experiences in identifying and referring individuals with mental health problems to the healthcare system, previous training on identifying such issues by healthcare professionals, and whether they had been consulted about mental health-related problems in the past week or month. This section of the survey was created by the authors.

The third section aimed to measure the religious leaders' perceptions of identifying and referring individuals with mental health problems to healthcare professionals. We used the Clergy's Perception of Mental Illness Survey instrument [13], which at the best of our knowledge has not been used in Peru before. This instrument presented four clinical cases involving depression, problematic drug use, problematic alcohol consumption, and schizophrenia. The participants were asked to assess if the individual in each case had a mental health problem, and then the type of problem (which of the four listed problems the case suffered from). They were also asked to rate the severity of the problem on a 5-point scale from 1 (not serious) to 5 (very serious). Finally, the participants, as leaders, were asked what decision they would make if the person with the problem sought help.

The Clergy's Perception of Mental Illness Survey instrument was translated into Spanish and adapted to the Peruvian cultural and linguistic context. The clinical cases are presented in English and Spanish in Table 2.

For all the survey, a clinical psychology teacher, a psychiatrist, and two epidemiologists validated its content by determining the relevance of the questions and the understandability of the text.

## Statistical analysis

We used measures of central tendency and dispersion for numerical variables and absolute and relative frequencies for categorical variables in the univariate analysis. We examined the relationship between the collected variables and the religious leaders' religious group, as well as the association between training from healthcare professionals on identifying mental health problems and the identification and referral of clinical cases to healthcare professionals. For these analysis, we used the Fisher and Kruskal-Wallis tests, considering a $p < 0.05$ as a statistically significant result. All statistical analyses were conducted using STATA version 15.

## Ethical considerations

Ethics approval was obtained from the local Institutional Review Board of the Universidad Católica los Angeles de Chimbote (ULADECH): IRB number 042-2018-CIEI-VI-ULADECH--Católica. We adhered to the ethical principles of human research [14].

All methods were carried out in accordance with relevant guidelines and regulations or declaration of Helsinki. Informed consent was obtained from all subjects. No participant was illiterate. Informed consent was obtained from all participants.

Before enrolling in the study, potential participants were approached at their places of worship and informed of the study's purpose. They were given written informed consent forms to read and were also given the opportunity to ask any questions before consenting to their

**Table 2. Clinical cases of the Clergy's Perception of Mental Illness Survey instrument.**

| Diagnosis | Clinical case in Spanish | Clinical case in English |
|---|---|---|
| Problematic drug use | Pedro es un adolescente que estudia secundaria. Hace un año, probó cocaína por primera vez con sus amigos en una fiesta. Durante los últimos meses, ha estado consumiendo cocaína en fiestas que duraban varios días. Ha perdido peso y ha experimentado escalofríos mientras festejaba. Pedro ha gastado todo su dinero en cocaína. Cuando sus amigos tratan de hablarle sobre los cambios en su conducta, él se enoja y se va. Sus amigos y familia han notado que algunas pertenencias han desaparecido y sospechan que Pedro les ha robado. Él ha tratado de dejar de consumir, pero no puede. Cada vez que trata de dejarlo se siente cansado, deprimido y no puede dormir. Hace un mes perdió su trabajo a medio tiempo por no asistir. | Pedro is a teenager who is in high school. A year ago, he tried cocaine for the first time with his friends at a party. In recent months, he has been using cocaine at parties that have lasted several days. He has lost weight and experienced chills while celebrating. Pedro has spent all his money on cocaine. When his friends try to talk to him about changes in his behavior, he gets angry and leaves. His friends and family have noticed that some belongings have disappeared, and they suspect that Pedro has stolen them. He has tried to stop consuming cocaine, but cannot. Every time he tries to quit, he feels tired, depressed, and cannot sleep. A month ago, he lost his part-time job for not going to work. |
| Depression | María es una mujer con educación secundaria. En las últimas dos semanas, María se ha sentido triste. Se levanta por las mañanas con una sensación de pesadez que la sigue todo el día. No disfruta las cosas que normalmente le agrada hacer. De hecho, nada le causa placer. Incluso cuando algo bueno le sucede, ella no se siente feliz. Se esfuerza por vivir el día a día, pero esto es muy difícil para ella. Las tareas más sencillas son muy difíciles de realizar. Le cuesta concentrarse. Se siente sin energía. A pesar de lo cansada que se siente, en las noches no consigue dormir. Ella siente que vale poco. Su familia ha notado que no es la misma que antes y que se ha distanciado desde hace un mes. No tiene ganas de hablar. | Maria is a woman with secondary education. In the past two weeks, Maria has felt sad. She gets up in the morning with a feeling of heaviness that follows her all day. She does not enjoy the things she normally likes to do. In fact, nothing gives her pleasure. Even when something good happens to her, she does not feel happy. She strives to live day to day, but this is very difficult for her. The simplest tasks are very difficult to perform. It is hard for her to concentrate. She feels she has no energy. Despite how tired she is feeling, she cannot sleep at night. She feels that she is unworthy. Her family has noticed that she is not the same as before and that she has distanced herself for a month. She does not want to talk. |
| Problematic alcohol consumption | Juan es un hombre con educación secundaria. Durante el último mes, ha empezado a beber más alcohol del habitual. De hecho, se ha dado cuenta que ahora necesita beber el doble para obtener el efecto deseado. Ha tratado de dejarlo muchas veces, pero no puede. Cada vez que ha tratado de dejarlo se agita, empieza a sudar y no puede dormir, por lo que vuelve a tomar alcohol. Su familia se queja de que está constantemente con resaca y se ha vuelto difícil confiar en él (hace planes y luego los cancela). | Juan is a man with secondary education. During the past month, he has started drinking more alcohol than usual. In fact, he has realized that he now needs to drink twice as much to get the desired effect. He has tried to quit many times, but he cannot. Every time he has tried to quit, he becomes agitated, starts to sweat, and cannot sleep. Therefore, he drinks alcohol again. His family complains that he has a constant hangover, and it has become difficult to trust him (he makes plans and then cancels them). |
| Schizophrenia | Ana es una mujer con educación superior. Hasta el año pasado su vida estaba bien. Pero las cosas han empezado a cambiar. Ella piensa que las personas a su alrededor hacen comentarios de desaprobación hacia su persona, que hablan a sus espaldas. Ana está convencida de que hay personas espiándola y que pueden oír lo que ella piensa. Ana ha perdido las ganas de participar en sus actividades cotidianas (trabajo/casa) y se ha retraído en su casa, la mayor parte del tiempo en su habitación. Ana escucha voces a pesar de que no hay nadie alrededor. Las voces le dicen qué hacer y qué pensar. Ella está viviendo así por seis meses. | Ana is a woman with higher education. Until last year, her life was fine. However, things have begun to change. She thinks that the people around her make disapproving comments about her and speak behind her back. Ana is convinced that there are people spying on her and that they can hear what she thinks. Ana has lost the desire to participate in her daily activities (at home or at work) and has started to stay at home, most of the time in her room. Ana hears voices although there is no one around. The voices tell her what to do and what to think. She has been living like this for six months. |

participation. All participants provided written consent by signing the informed consent documents, which were collected by the study team.

## Results

Of the five most common religious groups in Chimbote, the primary leader of the Jehovah Witnesses declined to participate in the study. However, leaders from the other four religious groups agreed to participate and provided lists of all their religious leaders. Out of a total of 196 potential participants, 109 participated in the study: 70 out of 150 Evangelicals, all 21 Mormons, 11 out of 18 Catholics, and all 7 Adventists agreed to participate.

While all the Mormon, Catholic, and Adventist leaders were male, 28.6% of the Evangelical leaders were female. The leaders' median age was 49 years (interquartile range = 40–60) and 92.7% were born in Peru. The results further revealed that 44% had a higher educational qualification. More specifically, 100%, 90%, and 24.3% of Adventists, Catholics, and Evangelicals had acquired a higher educational qualification. Moreover, 43.1% of the leaders had studied theology. While all the Catholic and Adventist leaders had studied theology, only 35.7% of the

Evangelical and 19.0% of the Mormon leaders had done so. Additionally, 27.5% had been religious leaders for 16 years or more. While none of the Mormon leaders had so much experience, 57.1% of the Adventist leaders did (Table 3).

While all Mormon, Catholic, and Adventist leaders were male, 28.6% of the Evangelical leaders were female. The median age of the leaders was 49 years and 92.7% were born in Peru. The results also showed that 44% had a higher education qualification (100%, 90%, and 24.3% of Adventists, Catholics, and Evangelicals). Additionally, 43.1% of the leaders had studied theology (all Catholic and Adventist leaders, while only 35.7% of Evangelical leaders and 19.0% of Mormon leaders). Moreover, 27.5% had been religious leaders for 16 years or more (none of the Mormon leaders had this level of experience, while 57.1% of the Adventist leaders did) (Table 3).

Concerning the enquiries the leaders received related to mental health problems, 38.5% and 50.5% reported that at least one of their parishioners had asked them for help during the previous week and the previous month, respectively. While the latter was the lowest among the Mormon leaders, it was the highest among their Adventist (71.4%) and Catholic (81.8%) counterparts. Most enquiries occurred before or after religious ceremonies (47.7%) and at the leaders' offices (14.7%). Irrespective of the religious group, the leaders noted the same frequencies. Although almost half of the leaders (48.6%) had advised at least one parishioner with mental health problems to go to a health facility, only 27.5% knew which healthcare centre they should be referring an individual to (Table 4). The results further revealed that between 22% and 30.3% of the leaders had received training on how to identify whether those who sought their advice were suffering from depression, problematic alcohol use, problematic drug use, and schizophrenia. Furthermore, between 88.1% and 89% stated that they would like to receive such training (Table 4).

Concerning the mental health problems, 38.5% and 50.5% of the leaders reported that at least one of their parishioners had asked for help during the previous week and month, respectively. While the latter was lowest among Mormon leaders (28.6%), it was highest among Adventist (71.4%) and Catholic (81.8%) leaders. Most enquiries occurred before or after religious ceremonies (47.7%) and at the leaders' offices (14.7%), regardless of religious group. Although almost

**Table 3. Characteristics of the religious leaders in Chimbote, Peru, 2019.**

| Characteristics | Total N = 109 | Catholic N = 11 | Evangelical N = 70 | Mormon N = 21 | Adventist N = 7 | P value * |
|---|---|---|---|---|---|---|
| | n (%) | n (%) | n (%) | n (%) | n (%) | |
| Male | 89 (81.7) | 11 (100.0) | 50 (71.4) | 21 (100.0) | 7 (100.0) | 0.002 |
| Age (years) ** | 49 (40–60) | 44 (38–48) | 53.5 (42–65) | 45 (36–53) | 51 (28–56) | 0.020 |
| Born in Peru | 101 (92.7) | 9 (81.8) | 67 (95.7) | 19 (90.5) | 6 (85.7) | 0.141 |
| Education | | | | | | <0.001 |
| 0–6 years | 28 (25.7) | 0 (0.0) | 28 (40.0) | 0 (0.0) | 0 (0.0) | |
| 7–11 years | 33 (30.3) | 1 (9.1) | 25 (35.7) | 7 (33.3) | 0 (0.0) | |
| 12 or more years | 48 (44.0) | 10 (90.9) | 17 (24.3) | 14 (66.7) | 7 (100.0) | |
| Studied theology | 47 (43.1) | 11 (100.0) | 25 (35.7) | 4 (19.0) | 7 (100.0) | <0.001 |
| Years of theology studies (among the 47 participants who studied theology) | 4 (3–6) | 7 (5–8) | 3 (2–4) | 2.5 (0.5–7) | 5 (5–5) | <0.001 |
| Length of time as a religious leader | | | | | | <0.001 |
| 1 month to 5 years | 43 (39.4) | 2 (18.2) | 21 (30.0) | 19 (90.5) | 1 (14.3) | |
| 6–15 years | 36 (33.0) | 5 (45.5) | 27 (38.6) | 2 (9.5) | 2 (28.6) | |
| 16–57 years | 30 (27.5) | 4 (36.4) | 22 (31.4) | 0 (0.0) | 4 (57.1) | |

* Fisher test was used for categorical variables, and Kruskal–Wallis test was employed for quantitative variables.

** Medium (interquartile range).

**Table 4. Past experiences related to the identification and referral of people with mental health problems to the healthcare system among religious leaders of Chimbote, Peru, 2019.**

| Variables | Total | Catholic | Evangelical | Mormon | Adventist | p-value (Fisher test) |
|---|---|---|---|---|---|---|
| | n = 109 | n = 11 | n = 70 | n = 21 | n = 7 | |
| | n (%) | n (%) | n (%) | n (%) | n (%) | |
| As a religious leader, has anyone asked you for help with mental health problems in the past week? | 42 (38.5) | 7 (63.6) | 27 (38.6) | 2 (9.5) | 6 (85.7) | <0.001 |
| As a religious leader, has anyone asked you for help with mental health problems in the past month? | 55 (50.5) | 9 (81.8) | 35 (50.0) | 6 (28.6) | 5 (71.4) | 0.022 |
| In which context do people usually ask you for help with mental health problems? (leaders were asked to give only one most common answer) | | | | | | <0.001 |
| Before or after the religious ceremonies | 52 (47.7) | 5 (45.5) | 42 (60.0) | 3 (14.3) | 2 (28.6) | |
| At the leader's office | 16 (14.7) | 2 (18.2) | 5 (7.1) | 7 (33.3) | 2 (28.6) | |
| Anytime | 13 (11.9) | 1 (9.1) | 8 (11.4) | 4 (19.0) | 0 (0.0) | |
| In visits and campaigns | 7 (6.4) | 0 (0.0) | 5 (7.1) | 2 (9.5) | 0 (0.0) | |
| The leader was not consulted regarding this topic | 7 (6.4) | 0 (0.0) | 3 (4.3) | 4 (19.0) | 0 (0.0) | |
| At the leader's house | 6 (5.5) | 0 (0.0) | 5 (7.1) | 0 (0.0) | 1 (14.3) | |
| Others | 8 (7.3) | 3 (27.3) | 2 (2.9) | 1 (4.8) | 2 (28.6) | |
| Leaders who said they know to which Chimbote health facility they can refer a person with mental health problems | 30 (27.5) | 5 (45.5) | 17 (24.3) | 6 (28.6) | 2 (28.6) | 0.523 |
| Leaders who have advised a person with mental health problems to go to a health facility at some time during their stay in Chimbote | 53 (48.6) | 8 (72.7) | 29 (41.4) | 11 (52.4) | 5 (71.4) | 0.147 |
| Leaders who have received any training from healthcare personnel to: | | | | | | |
| Identify a person with depression | 33 (30.3) | 8 (72.7) | 15 (21.4) | 4 (19.0) | 6 (85.7) | <0.001 |
| Identify a person with problems with alcohol or drug consumption | 29 (26.6) | 8 (72.7) | 14 (20.0) | 3 (14.3) | 4 (57.1) | 0.001 |
| Identify a person suffering from schizophrenia, psychosis, or hallucinations | 24 (22.0) | 6 (54.5) | 12 (17.1) | 3 (14.3) | 3 (42.9) | 0.020 |
| Leaders who would like to receive training provided by healthcare personnel to: | | | | | | |
| Recognize depression | 96 (88.1) | 11 (100.0) | 58 (82.9) | 21 (100.0) | 6 (85.7) | 0.087 |
| Recognize problems with alcohol or drug consumption | 96 (88.1) | 11 (100.0) | 57 (81.4) | 21 (100.0) | 7 (100.0) | 0.042 |
| Recognize schizophrenia, psychosis, or hallucinations | 97 (89.0) | 11 (100.0) | 58 (82.9) | 21 (100.0) | 7 (100.0) | 0.084 |
| Leaders who agree or strongly agree with the following statement: "If any of your parishioners have a mental health problem, would you send him/her to a healthcare facility?" | 92 (84.4) | 11 (100.0) | 53 (75.7) | 21 (100.0) | 7 (100.0) | 0.011 |

half of the leaders (48.6%) had advised at least one parishioner with mental health problems to go to a healthcare facility, only 27.5% affirmed to know which healthcare center they should refer an individual to. The results also showed that between 22% and 30.3% of the leaders had received training on how to identify whether those seeking their advice were suffering from depression, problematic alcohol use, problematic drug use, or schizophrenia. Furthermore, between 88.1% and 89% stated that they would like to receive such training (Table 4).

While all Catholic, Mormon, and Adventist leaders stated that they would send their parishioners with mental health problems to a healthcare facility, only 75.7% of Evangelical leaders did so (Table 4).

The assessment of the leaders' perceptions of the four clinical cases showed that 85.9% correctly identified depression, 97.6% determined problematic alcohol consumption, and 93.7% recognized problematic drug use. However, only 62.0% correctly identified schizophrenia. The percentage of leaders who stated that they would refer each of these cases to a healthcare professional ranged from 79.8% to 82.6%. However, all those who declared they would not refer these cases were Evangelicals (Table 5).

Leaders who had received training to identify various mental health problems had a higher rate of recognizing such in the clinical cases, although the results were only statistically

**Table 5. Perceptions of religious leaders in Chimbote, Peru, of the presented clinical cases, 2019 (n = 109).**

| Variables | Case 1 (Problematic drug use) | Case 2 (Depression) | Case 3 (Problematic alcohol consumption) | Case 4 (Schizophrenia or psychosis) |
|---|---|---|---|---|
| | n (%) | n (%) | n (%) | n (%) |
| In your opinion, the person of the clinical case has a medical or mental health problem | 79 (72.5) | 77 (70.6) | 83 (76.2) | 79 (72.5) |
| Among those who mentioned that the clinical case had a medical or mental health problem: | | | | |
| Correctly answered the diagnosis of the clinical case | 74 (93.7) | 67 (85.9) | 81 (97.6) | 49 (62.0) |
| How serious do you think the problem is? [from 1 (not serious) to 5 (very serious)]: Mean ± standard deviation | 4.2 ± 0.7 | 3.8 ± 0.8 | 4.0 ± 0.7 | 4.2 ± 0.9 |
| If the person of the clinical case came to you to ask for help, you. . . | | | | |
| would consider that he/she does not require help | 0 (0.0) | 0 (0.0) | 0 (0.0) | 0 (0.0) |
| would help him/her without sending him/her to a health professional | 19 (17.4) | 22 (20.2) | 20 (18.4) | 19 (17.4) |
| would help him/her and also would send him/her to a healthcare professional | 90 (82.6) | 84 (77.1) | 87 (79.8) | 88 (80.7) |
| would only send him/her to a healthcare professional | 0 (0.0) | 3 (2.8) | 2 (1.8) | 2 (1.8) |
| Participants who answered "would send him/her to a healthcare professional," by religious group: | | | | |
| Catholic (n = 11) | 11/11 (100.0) | 11/11 (100.0) | 11/11 (100.0) | 11/11 (100.0) |
| Evangelical (n = 70) | 51/70 (72.9) | 48/70 (68.6) | 50/70 (71.4) | 51/70 (72.9) |
| Mormon (n = 21) | 21/21 (100.0) | 21/21 (100.0) | 21/21 (100.0) | 21/21 (100.0) |
| Adventist (n = 7) | 7/7 (100.0) | 7/7 (100.0) | 7/7 (100.0) | 7/7 (100.0) |

significant for depression and problematic alcohol consumption. However, there were no statistically significant associations between having received training and being willing to refer the respective clinical case to a healthcare professional (Table 6).

## Discussion

In this study, approximately half of the religious leaders reported that their parishioners had asked them for help with mental health problems in the previous month. This finding is

**Table 6. Association between having received training for identifying or willingness to refer the clinical cases to healthcare system among religious leaders.**

| Received training to identify the following mental health problems | Identified the clinical case | | | Would refer the clinical case | | |
|---|---|---|---|---|---|---|
| | No | Yes | p value | No | Yes | p value |
| Depression | | | 0.018 | | | 0.583 |
| No | 35 (46.1) | 41 (53.9) | | 2 (2.6) | 74 (97.4) | |
| Yes | 7 (21.2) | 26 (78.8) | | 2 (6.1) | 31 (93.9) | |
| Problematic drug use | | | 0.165 | | | 0.776 |
| No | 29 (36.3) | 51 (63.8) | | 15 (18.8) | 65 (81.3) | |
| Yes | 6 (20.7) | 23 (79.3) | | 4 (13.8) | 25 (86.2) | |
| Problematic alcohol consumption | | | 0.028 | | | 0.422 |
| No | 25 (31.3) | 55 (68.8) | | 18 (22.5) | 62 (77.5) | |
| Yes | 3 (10.3) | 26 (89.7) | | 4 (13.8) | 25 (86.2) | |
| Schizophrenia | | | 0.064 | | | 0.393 |
| No | 51 (60.0) | 34 (40.0) | | 19 (22.4) | 66 (77.6) | |
| Yes | 9 (37.5) | 15 (62.5) | | 3 (12.5) | 21 (87.5) | |

consistent with studies conducted in the United States and Nigeria involving Catholics and Mormons [4,7,8,13]. Some parishioners may consider their mental health problems to have a spiritual origin [15] because leaders discuss mental health during religious ceremonies and when they preach [16]. Furthermore, they perceive their leaders as advisors with whom they can confidentially discuss these issues [17].

While Catholic and Adventist leaders received the most enquiries about mental health problems, Mormon leaders received the least. This may be related to differences in the frequency and type of interaction leaders have with their parishioners about what constitutes a mental health problem. Further research is recommended to confirm and deepen these results.

More than 85% of the religious leaders identified depression and problematic alcohol and/or drug use correctly when asked to identify the clinical cases. However, less than two-thirds were able to identify schizophrenia. This factor is in agree with studies that found that more religious leaders identified depression correctly than schizophrenia [8,17]. This aspect could be explained by the lower prevalence of schizophrenia in comparison to depression and alcohol consumption [18–20]. Thus, awareness of schizophrenia may be lower. However, this lower recognition of the schizophrenia case did not appear to affect the religious leaders' identification of the case as a mental health problem and their decision to refer the case to a healthcare professional. The frequency thereof was similar to that of the other cases.

More than 85% of the religious leaders correctly identified the clinical cases of depression and problematic alcohol and/or drug use. However, less than two-thirds were able to identify schizophrenia. This is consistent with studies that found that more religious leaders correctly identified depression than schizophrenia [8,17]. This may be due to the lower prevalence of schizophrenia compared to depression and alcohol consumption [18–20], leading to lower awareness of schizophrenia. However, this lower recognition of schizophrenia did not appear to affect the religious leaders' identification of the case as a mental health problem or their decision to refer the case to a healthcare professional. The frequency of referral was similar to that of the other cases.

Most of the religious leaders wanted to refer parishioners to a healthcare professional and less than 20% wanted to help these parishioners themselves. These results suggest that it may be feasible to create partnerships between religious leaders and healthcare professionals in an effort to establish a referral pathway for individuals with mental health problems.

Approximately one-quarter of the Evangelical leaders were reluctant to refer parishioners with mental health problems to a healthcare professional. This is consistent with a study of 107 religious' leaders in Nigeria that found that 35.6% and 42.4% of Evangelical leaders, 38.9% and 22.2% of Muslim leaders, and 22.6% and 16.7% of Orthodox leaders did not refer individuals with depression and schizophrenia, respectively, to a healthcare professional [8]. Accordingly, a study of 230 Evangelical leaders from various countries gathered in the United States found that they placed more value on spiritual discipline and a Bible-based interpretation of depression than on psychological or medical treatment [21]. Similarly, a survey of 20 Evangelical leaders from Ghana with over 20 years of leadership experience found that all the leaders agreed that the origin of depression is sin or estrangement from God [22]. However, these interpretations should be complemented with qualitative assessments.

Most religious leaders in this study expressed interest in receiving training to identify and refer individuals with mental health problems to the healthcare system. Educational programs may improve religious leaders' perceptions and practices, enabling them to recognize the importance of mental health cases and increase the number of referrals to the healthcare system [23,24]. Based on our results, it is recommended that the training should include practical information on where to refer individuals with mental health problems.

### Limitations and strengths

This study has some limitations that should be considered. First, there is the possibility of selection bias due to low participation rates among Evangelical (47%) and Catholic (61%) leaders, which may have led to a lack of representation from leaders who have less contact with parishioners. Second, social desirability bias may have affected the responses of the leaders, who knew they were participating in a study and may have provided responses they thought the interviewer wanted to hear. However, this bias may have been reduced by emphasizing the anonymity of the survey and the desire to collect true perceptions before obtaining informed consent. Third, our study primarily explores the perspectives and needs of religious leaders within a specific context, which necessitates caution when extrapolating our results; however, it contributes to the broader conversation about the intersection of religion and mental health. Finally, the study did not explore other factors, as the etiology of mental health problems as perceived by religious leaders, which may have provided additional insight into their responses related to the clinical cases.

Despite these limitations, to our best knowledge, this is the first study to examine the experiences and perceptions of religious leaders from different religious groups in Latin America related to identifying and referring individuals with possible mental health problems to the healthcare system. The results of this study provide a foundation for future research on the collaborative work between religious leaders and healthcare providers to improve access to mental health services in the community.

## Conclusion

Our study found that religious leaders were frequently approached by their parishioners for help with mental health issues. Although most leaders expressed a desire for training, less than a third had received any. However, most leaders were able to correctly identify clinical cases of mental health problems and expressed willingness to refer individuals to healthcare professionals. Notably, none of the Evangelical leaders intended to refer individuals to healthcare professionals. These results suggest that collaboration with religious leaders is feasible and could help identify and refer individuals with mental health issues, but further research is needed to identify effective methods for each religious group.

## Supporting information

**S1 Data.**
(XLSX)

## Acknowledgments

We thank Vania Dioses Sifuentes and David Anthony Cano Horna for their support in conducting the fieldwork.

## Author Contributions

**Conceptualization:** Julio Cjuno, Jessica Hanae Zafra-Tanaka, Teresa del Pilar García García, Alvaro Taype-Rondan.

**Formal analysis:** Alvaro Taype-Rondan.

**Investigation:** Jessica Hanae Zafra-Tanaka.

**Methodology:** Julio Cjuno, Jessica Hanae Zafra-Tanaka, Teresa del Pilar García García, Alvaro Taype-Rondan.

**Writing – original draft:** Julio Cjuno, Jessica Hanae Zafra-Tanaka, Alvaro Taype-Rondan.

**Writing – review & editing:** Julio Cjuno, Jessica Hanae Zafra-Tanaka, Teresa del Pilar García García.

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
