## [Decision Letter · Decision Letter 0]

21 Sep 2023

PONE-D-23-10333Religious leaders’ perceptions of the identification and referral of people with mental health problems in a Peruvian cityPLOS ONE

Dear Dr. Taype-Rondan,

Thank you for submitting your manuscript to PLOS ONE. After careful consideration, we feel that it has merit but does not fully meet PLOS ONE’s publication criteria as it currently stands. Therefore, we invite you to submit a revised version of the manuscript that addresses the points raised during the review process.

The author should carefully consider the first reviewer's criticisms and incorporate their suggestions into the presentation of results. The presentation of results beyond descriptive data should be more detailed and explained in the body of the text. The inferential analysis presented at the moment makes the contribution much less relevant to the field than a more in-depth analysis of the valuable data collected. To enhance the clarity of the writing, the authors could consider using visual aids to illustrate key findings. This would help readers to better understand the data and the significance of the results.

We look forward to receiving your revised manuscript.

Kind regards,

Ivan Filipe de Almeida Lopes Fernandes, Ph.D.

Academic Editor

PLOS ONE

Journal Requirements:

Reviewers' comments:

Reviewer's Responses to Questions

**Comments to the Author**

1. Is the manuscript technically sound, and do the data support the conclusions?

Reviewer #1: No

Reviewer #2: Yes

2. Has the statistical analysis been performed appropriately and rigorously? 

Reviewer #1: I Don't Know

Reviewer #2: Yes

3. Have the authors made all data underlying the findings in their manuscript fully available?

Reviewer #1: Yes

Reviewer #2: Yes

4. Is the manuscript presented in an intelligible fashion and written in standard English?

Reviewer #1: No

Reviewer #2: Yes

5. Review Comments to the Author

Reviewer #1: The relevance of the paper's aim, i.e., to understand religious leaders' perceptions of mental illness may be contextual. That is, it is not clear why it would be incumbent upon religious leaders to be competent in identifying mental health issues. Although the authors have stated that mental illness is a global problem, the role of the addressed population of the study in this issue may be of local importance at most. In other words, no justification was provided for selecting parishioners as legitimate diagnosticians of mental illness. There is no possible explanation in the text for attributing this role to them in the analyzed context. In any case, even if there is, this can seriously compromise the possibility of generalization or even extension of the conclusions beyond the context itself.

Regarding the sample, there is a significant discrepancy between the number of leaders of each religion included in the research. There was also no relationship between the percentage of believers per religion and the number of leaders. Finally, considering that the religions addressed have very distinct formation processes, and may attribute people from different social and economic classes, it seems to me that the sample was not well designed.

Finally, and perhaps most relevant, the research instrument (survey) is especially epistemically fragile. The complexity of the selected mental illnesses (drug use, depression, problematic alcohol consumption, and schizophrenia) is not reducible to what are, in my opinion, incorrectly called "clinical cases". The very short text is not, in my evaluation, appropriate for assessing the technical competence of religious leaders.

In summary, there are many weaknesses in the paper presented: it is not clear why religious leaders should be responsible for referring people to health services; the sample does not seem consistently constituted, and the survey instrument is clearly inappropriate. Taken together, these factors lead me not to suggest its publication.

Reviewer #2: This is a well planned and done piece of work with implications for the mental health assistance in median and low income countries.

Statistics although well done could be improved. A correlation between education level (theology for instance) with diagnosis, informations about mental illness and what to do with the ill parishioners might strenght the necessity of training.

If there is information available about demographic characterictics of the priests who refused to participate, this will say much about the representiveness of the sample.

6. PLOS authors have the option to publish the peer review history of their article (what does this mean?). If published, this will include your full peer review and any attached files.

Reviewer #1: No

Reviewer #2: No

---

## [Author Response · Author response to Decision Letter 0]

9 Nov 2023

Dear editor

We express our gratitude for the insightful commentaries provided by the reviewer. In response, we will address each of their points in detail:

Reviewer #1: The relevance of the paper's aim, i.e., to understand religious leaders' perceptions of mental illness may be contextual. That is, it is not clear why it would be incumbent upon religious leaders to be competent in identifying mental health issues. Although the authors have stated that mental illness is a global problem, the role of the addressed population of the study in this issue may be of local importance at most. In other words, no justification was provided for selecting parishioners as legitimate diagnosticians of mental illness. There is no possible explanation in the text for attributing this role to them in the analyzed context. In any case, even if there is, this can seriously compromise the possibility of generalization or even extension of the conclusions beyond the context itself.

Answer: We appreciate your thoughtful review of our paper and your valuable insights. You raise a valid point regarding the contextual relevance of our paper's aim. We acknowledge that the significance of understanding religious leaders' perceptions of mental illness may indeed vary depending on the local context.

To clarify this, we have added the following text into the introduction section: “This reliance on religious leaders underscores the potential for collaboration between healthcare services and religious institutions to equip the latter with the skills needed to recognize and refer parishioners with suspected mental health issues to appropriate healthcare services. Studies in the United States (4,7) and Nigeria (8) have found that although religious leaders often receive inquiries about mental health problems from their parishioners, they may not always possess the necessary expertise to accurately recognize these issues and often wish to receive training to do so.”

Also, in the Limitations section, we added the following text: “Third, our study primarily explores the perspectives and needs of religious leaders within a specific context, which necessitates caution when extrapolating our results; however, it contributes to the broader conversation about the intersection of religion and mental health.”

Also, while we acknowledge that our study focuses on a local population, we firmly believe that this should not impede our contribution from being published in an international journal. Similar studies conducted elsewhere have been successfully published in such journals, and we are confident that our research adds valuable insights to the broader conversation on this subject: https://pubmed.ncbi.nlm.nih.gov/?term=religious+leaders+mental+health

Review 2: Regarding the sample, there is a significant discrepancy between the number of leaders of each religion included in the research. There was also no relationship between the percentage of believers per religion and the number of leaders. Finally, considering that the religions addressed have very distinct formation processes, and may attribute people from different social and economic classes, it seems to me that the sample was not well designed.

Answer: Thank you for your commentary. Our intention in this study was to include all possible religious leaders from the different religions represented in the community. We aimed to capture a comprehensive view of the religious leadership landscape rather than making sample size decisions based on the proportion of believers within each religion. By doing so, we aimed to provide a nuanced and balanced representation of the religious leaders in the community.

Finally, and perhaps most relevant, the research instrument (survey) is especially epistemically fragile. The complexity of the selected mental illnesses (drug use, depression, problematic alcohol consumption, and schizophrenia) is not reducible to what are, in my opinion, incorrectly called "clinical cases". The very short text is not, in my evaluation, appropriate for assessing the technical competence of religious leaders.

Answer: Our survey instrument employed simple questions to ensure accessibility for a diverse sample (religious leaders of different backgrounds and religions). Moreover, the instrument used (Clergy's Perception of Mental Illness Survey instrument) has been used in previous study, which allows comparability (as detailed in the “Instrument and variables” subheading of the Methods section).

Reviewer #2: This is a well planned and done piece of work with implications for the mental health assistance in median and low income countries.

Answer: Thank you very much for your kind commentary.

Statistics although well done could be improved. A correlation between education level (theology for instance) with diagnosis, informations about mental illness and what to do with the ill parishioners might strenght the necessity of training.

Answer: We appreciate your suggestion to explore the correlation between education level and aspects related to diagnosis, information about mental illness. While we recognize the importance of this analysis, it's worth noting that our study did find a strong association between education and religion (see Table 3). Specifically, Catholics and Adventists in our sample tended to have more years of education and theological training. Given this association, assessing the relationship between education and knowledge about mental health could potentially be confounded by religious affiliation.

If there is information available about demographic characterictics of the priests who refused to participate, this will say much about the representiveness of the sample.

Answer: Thank you very much for your commentary. Regretfully, we did not have access to such information.

---

## [Decision Letter · Decision Letter 1]

21 Feb 2024

Religious leaders’ perceptions of the identification and referral of people with mental health problems in a Peruvian city

PONE-D-23-10333R1

Dear Dr. Taype-Rondan,

We’re pleased to inform you that your manuscript has been judged scientifically suitable for publication and will be formally accepted for publication once it meets all outstanding technical requirements.

Kind regards,

Ivan Filipe de Almeida Lopes Fernandes, Ph.D.

Academic Editor

PLOS ONE

Additional Editor Comments (optional):

Reviewers' comments:

Reviewer's Responses to Questions

**Comments to the Author**

1. If the authors have adequately addressed your comments raised in a previous round of review and you feel that this manuscript is now acceptable for publication, you may indicate that here to bypass the “Comments to the Author” section, enter your conflict of interest statement in the “Confidential to Editor” section, and submit your "Accept" recommendation.

Reviewer #2: All comments have been addressed

2. Is the manuscript technically sound, and do the data support the conclusions?

Reviewer #2: Yes

3. Has the statistical analysis been performed appropriately and rigorously? 

Reviewer #2: Yes

4. Have the authors made all data underlying the findings in their manuscript fully available?

Reviewer #2: Yes

5. Is the manuscript presented in an intelligible fashion and written in standard English?

Reviewer #2: Yes

6. Review Comments to the Author

Reviewer #2: This is a relevant article that might stimulate valuable better training to priests and religious leaders. They are in touch with people and can help to detect mental health problems, address them in a proper way and give correct advice.

7. PLOS authors have the option to publish the peer review history of their article (what does this mean?). If published, this will include your full peer review and any attached files.

Reviewer #2: No

---

## [Editor Report · Acceptance letter]

27 Feb 2024

PONE-D-23-10333R1 

PLOS ONE

Dear Dr. Taype-Rondan, 

I'm pleased to inform you that your manuscript has been deemed suitable for publication in PLOS ONE. Congratulations! Your manuscript is now being handed over to our production team.

Kind regards, 

on behalf of

Dr. Ivan Filipe de Almeida Lopes Fernandes 

Academic Editor

PLOS ONE